# Breastfeeding and Obstetric Violence during the SARS-CoV-2 Pandemic in Spain: Maternal Perceptions

**DOI:** 10.3390/ijerph192315737

**Published:** 2022-11-26

**Authors:** Desirée Mena-Tudela, Susana Iglesias-Casas, Agueda Cervera-Gasch, Laura Andreu-Pejó, Victor Manuel González-Chordá, María Jesús Valero-Chillerón

**Affiliations:** 1Department of Nursing, Faculty of Health Sciences, Universitat Jaume I, Avda. Sos I Baynat s/n, 12071 Castellón, Spain; 2Department of Obstetrics, Hospital do Salnés, Villgarcía de Aurousa, 36619 Pontevendra, Spain

**Keywords:** breastfeeding, obstetric violence, violence against women, women’s health

## Abstract

Catalan legislation, a pioneer in Europe, has defined obstetric violence (OV) as “preventing or hindering access to truthful information, necessary for autonomous and informed decision-making”. The definition also states that OV can affect physical and mental health, as well as sexual and reproductive health. Some authors have expressed concern about an increase in OV during the SARS-CoV-2 pandemic. During the pandemic, recommendations were also openly offered on the non-establishment and/or early abandonment of breastfeeding without scientific evidence to support them. Experiencing a traumatic childbirth can influence breastfeeding outcomes. Here, we conducted a cross-sectional study using a self-administered online questionnaire. The sample consisted of women who gave birth in Spain between March 2020 and April 2021. The mean age was 34.41 (±4.23) years. Of the women, 73% were employed, 78.2% had a university education, and almost all were Caucasian. Among the subjects, 3.3% were diagnosed with SARS-CoV-2 during pregnancy and 1% were diagnosed during delivery. Some of the women (1.6%) were advised to stop breastfeeding in order to be vaccinated. Women diagnosed with SARS-CoV-2 during delivery (*p* = 0.048), belonging to a low social class (*p* = 0.031), with secondary education (*p* = 0.029), or who suffered obstetric violence (*p* < 0.001) perceived less support and that the health care providers were less inclined to resolve doubts and difficulties about breastfeeding. Breastfeeding has been significantly challenged during the pandemic. In addition to all the variables to be considered that make breastfeeding support difficult, we now probably need to add SARS-CoV-2 diagnosis and OV.

## 1. Introduction

The United Nations Population Fund (UNFPA) has acknowledged that there is a lack of consensus in health care settings on how to define and measure violence against women during childbirth [1]. Despite this, there are various definitions of obstetric violence around the world. One of the first legal definitions is found in Venezuela’s Organic Law on Women’s Right to Live a Life Free of Violence, which defines obstetric violence as “the appropriation of women’s bodies and reproductive processes by health personnel, which is expressed in a dehumanising treatment, in an abuse of medicalisation and pathologisation of natural processes, bringing with it a loss of autonomy and the capacity to decide freely about their bodies and sexuality, negatively impacting women’s quality of life” [2]. In Europe, for some years, obstetric violence has been an increasingly debated issue, mainly promoted by organisations and social movements in order to defend human rights [3]. Recently, in Spain, Catalan legislation has been a pioneer in Europe in defining and including obstetric violence in its legislation, defining it as “preventing or hindering access to truthful information, necessary for autonomous and informed decision-making. It can affect […] physical and mental health, including sexual and reproductive health, and can prevent or hinder women from making decisions about their sexual practices and preferences, and about their reproduction” [4]. Despite the lack of consensus and the different definitions of obstetric violence, all the definitions seem to have two common denominators: the loss of autonomy on the part of the woman and the harm caused.

In addition, some authors have expressed concern about an increase in obstetric violence during the pandemic [5]. Breastfeeding is rarely considered in the scientific literature as a sexual and reproductive right, and it is fundamentally neglected from a gender perspective. Thus, during the SARS-CoV-2 pandemic, recommendations on the non-establishment and/or early abandonment of breastfeeding were openly offered without scientific evidence to justify such decisions [6]. Since the onset of the well-known COVID-19 pandemic, agencies such as the World Health Organisation have supported the introduction and maintenance of breastfeeding [7] and, subsequently, demonstrated the robust immune response of the female body to the SARS-CoV-2 virus by the passage of antibodies through breast milk [8]. Although the scientific literature has not reported that pregnant and lactating women are at an increased health risk due to SARS-CoV-2 infection [6], changes in health care, social policy, and social circumstances have led to a change in the care of women in childbirth across Europe [9]. It should be noted that the literature shows that the experience of a traumatic event during childbirth influences breastfeeding outcomes [10] and can be experienced as a dichotomy in which breastfeeding is either a burden and a constant reminder of the trauma or a redemptive experience [11], although most women who have had a traumatic childbirth experience problems with breastfeeding [12]. These women require extra support to continue their decision to breastfeed [10]. During the SARS-CoV-2 pandemic, it has been shown that experiencing a traumatic childbirth may have changed the mother’s decision to breastfeed [13].

Thus, the aim of this study was to analyse women’s perceptions of obstetric violence related to breastfeeding support in Spain during the SARS-CoV-2 pandemic.

## 2. Materials and Methods

A cross-sectional study was conducted by means of a voluntary and anonymous online questionnaire using the Google Forms platform and distributed via social networks. The study was designed in accordance with the principles of the Declaration of Helsinki and under the Organic Law 03/2018 on Personal Data Protection and Guarantee of Digital Rights and received ethical approval from the Universitat Jaume I (CD/06/2021). The first question on the online survey asked women to confirm consent for their participation in the study.

The sample was selected non-randomly using non-probability sampling. Participants were eligible for the survey if they had given birth in Spain from March 2020 to April 2021. For the analysis of the results, questionnaires that did not correctly fill in the date of childbirth were excluded.

An online survey was developed for this study and hosted on the secure Google Forms platform. A modified version of the survey used for the measurement of obstetric violence in Spain was used [14,15,16]. The research team decided to add sociodemographic variables, obstetric variables, and variables related to SARS-CoV-2 to this tool. Some variables were also modified to take into consideration new possible interventions related to the pandemic, such as giving birth with a mask on or lack of maternity classes. Similarly, other variables were added, such as whether a woman thought that obstetric violence was justified by the pandemic. Breastfeeding-related variables explored the perception of support, the offer of formula feeding, contact with support groups, cessation of breastfeeding for vaccination, and women’s feelings. All of these variables could be answered with: Yes, No, Don’t Know, No Answer.

The survey was pilot tested (*n* = 20) to check the clarity and comprehension of each variable. Following the test, minor modifications were made to the sentence structure. The possibility that women would find it difficult to respond to the survey was taken into consideration, and therefore, the survey was kept as brief as possible. In the pilot test, it was observed that about 10 min were sufficient to answer the survey.

Women were recruited through social networks (Facebook, Twitter, and Instagram). Activist networks such as the Observatorio de Violencia Obstétrica en España and El Parto es Nuestro also participated in the dissemination of the study. In this case, an anonymous generic link was distributed.

A descriptive analysis was performed for all variables using either frequency and percentage or mean and standard deviation, according to the nature of the variable. For the SARS-CoV-2 diagnostic variables, a bivariate analysis was carried out using the chi-square test through contingency tables or one-factor ANOVA, according to the nature of the variables. Data were processed using the Statistical Package for the Social Sciences (SPSS) v. 25 (IBM, Armonk, NK, United States of America). Graphs were created in Excel spreadsheets. A statistical significance level of *p* < 0.05 was assumed.

## 3. Results

A total of 6270 responses were obtained. However, 210 (3.34%) questionnaires were eliminated due to the fact that the childbirth date was not filled in correctly because they had given birth before the pandemic. Therefore, the final sample consisted of 6060 questionnaires.

The sociodemographic characteristics of the women who participated in the study included a mean age of 34.41 (±4.23) years, 73% (*n* = 4423) were employed, 78.2% (*n* = 4736) had a university education, 93.5% (*n* = 5664) belonged to the middle social class, and 97.1% (*n* = 5887) were white or Caucasian. It should be noted that 3.3% (*n* = 200) of the women were diagnosed with COVID-19 during pregnancy and 1% (*n* = 63) were diagnosed during childbirth. All the sociodemographic variables are presented in Table 1.

Regarding the variables related to breastfeeding, 27.5% (*n* = 1590) of the women stated that they did not feel supported in their decisions about feeding and/or caring for their baby; 27.2% (*n* = 1536) were offered formula milk; 76.3% (*n* = 4315) were not offered contact with breastfeeding support groups; 32.5% (*n* = 1819) did not feel supported to resolve doubts or difficulties; and, despite no evidence being available, 1.6% (*n* = 88) of the women were advised to give up breastfeeding in order to get vaccinated. In addition, 26% (*n* = 1578) of the women surveyed reported experiencing obstetric violence during care throughout the SARS-CoV-2 pandemic (Table 2).

In terms of support and resolution of doubts and difficulties related to breastfeeding, it was observed that women with a positive diagnosis for SARS-CoV-2 at childbirth (46.8%, *n* = 29, *p* = 0.048); women of a low self-perceived social class (40.0%, *n* = 92, *p* = 0.031); women with secondary education (35.7%, *n* = 411, *p* = 0.029); and women who had experienced obstetric violence during their care (46.4%, *n* = 665, *p* < 0.001) tended to receive less support and help to resolve doubts and difficulties related to breastfeeding. Table 3 and Figure 1 show a comparative analysis between the variables collected related to breastfeeding support and obstetric violence. As can be seen in Figure 1, there are statistically significant differences in women’s perceptions of breastfeeding support depending on whether they have experienced obstetric violence. Thus, women who reported having suffered obstetric violence received less effective support (*p* < 0.001), were not supported in their decisions (*p* < 0.001), were offered more formula milk (*p* < 0.001), and were not put in contact with support groups (*p* < 0.001).

Table 4 shows the comparative analysis of variables related to breastfeeding and maternal SARS-CoV-2 diagnosis. Notably, women who were diagnosed with SARS-CoV-2 during childbirth received less support and had fewer questions or difficulties resolved, with statistically significant differences (*p* = 0.048).

## 4. Discussion

Women and their desire to initiate and/or maintain breastfeeding have faced a major challenge during the pandemic [6,17]. As the results of this study show, breastfeeding was compromised during the SARS-CoV-2 pandemic because, despite international recommendations to support breastfeeding during the pandemic [7], many women received conflicting indications that, in some cases, advocated the abandonment of breastfeeding. The robust immunological response of the female body to COVID-19 [8] and to the passage of SARS-CoV-2 antibodies through breast milk with vaccination have been demonstrated in the scientific literature [18].

One in four women in this study reported experiencing obstetric violence. There are middle-range theories whose axioms provide that traumatic birth can negatively affect a mother’s breastfeeding experience [10]. Axioms are assumptions that are considered researchable, as a whole, and provide a context for analytical processes. The relationship between obstetric violence and breastfeeding needs to be explored in depth. Earlier studies provide such harrowing accounts as: “Breastfeeding was just one of the many things I did while remaining totally detached from my baby” [11], which expresses the ordeal and impact of childbirth trauma on breastfeeding. Women who have experienced obstetric violence may require extra support for breastfeeding [10]. With all of the above, one in four women were offered formula milk during their hospital stay and one in three women did not feel effectively supported in their doubts or difficulties. The scientific literature reports, on the one hand, that the use of dummies and/or bottles may be associated with unfavourable breastfeeding behaviours, with bottle use in particular, and may affect important areas such as breastfeeding position, sucking pattern, infant response, breast anatomy, and the mother–child relationship [19]. On the other hand, it is well known that support for breastfeeding women is crucial for the optimal establishment and maintenance of breastfeeding, increasing their self-esteem and confidence and reducing social isolation [20]. Future research needs to delve deeper into the complex dynamics of obstetric violence on women’s breastfeeding experiences. It should be highlighted that two out of three women indicated that the hospital setting was the place where they did not feel supported regarding their decisions related to the feeding of their child, although, paradoxically, it was the primary care setting that received the most pressure in terms of care during the SARS-CoV-2 pandemic, as well as the least human and economic resources to reinforce the pandemic’s containment in Spain [21].

Knowing the profiles of women who feel or receive less support is important in order to be able to focus some finite efforts, such as human and material resources. Thus, in the present study, women of low social class, with a secondary education, with a positive SARS-CoV-2 diagnosis during childbirth, and those who had perceived obstetric violence in their care received the least support. Changes in breastfeeding plans have also been observed during the SARS-CoV-2 pandemic after a traumatic childbirth [13]. Some authors have already pointed out that social and cultural factors need to be taken more into consideration in order to understand the extent to which they may affect breastfeeding practices [22]. Further studies are needed to explore this important issue.

Other factors related to the lack of support and the potential impact on breastfeeding rates may include, as discussed above, a maternal SARS-CoV-2 diagnosis and the perception of obstetric violence in obstetric care. Regarding the diagnosis and impact of SARS-CoV-2, direct results on breastfeeding rates have already been seen, with all cases showing a decrease in breastfeeding rates [23] despite efforts to put in place protocols to contain this situation [24] or to implement baby-friendly hospital initiative (BFHI) protocols [25]. It should be noted that none of the variables analysed showed statistically significant differences according to maternal diagnosis for SARS-CoV-2, except for effective support with doubts or difficulties, in line with other international studies [26,27]. This scenario was also observed in Spain, and some studies suggest that this lack of support presence was replaced by online tools [28,29]. These aspects also constitute an interesting line of research. Regarding obstetric violence, it should be highlighted that, despite the fact that breastfeeding is part of a woman’s reproductive life, it has not been included in the research and in subsequent reviews that have been carried out on this subject [30,31,32], even though, as seen in our results, women who have experienced obstetric violence in their care report worrying figures regarding breastfeeding-related care such as less support, greater offering of formula feeding, less contact with support groups in the area, or greater recommendations to abandon breastfeeding to get vaccinated against SARS-CoV-2, facts that have also been reported by other studies [17,33]. In this sense, it is also necessary to highlight that women who answered “don’t know” to whether they had suffered obstetric violence in their care obtained very similar results in all variables related to breastfeeding as women who answered “yes”. These findings invite us to reflect on the lack of visibility of this type of violence, its normalisation, and the structural problem it represents both for women and health professionals [30,34,35,36,37]. Determining the relationship between obstetric violence and breastfeeding can also be used to formulate concrete policies or guidelines for action to address this major problem. In this regard, more research on the possible relationship between obstetric violence and breastfeeding is urgently needed.

Finally, it should be noted that this study is not without limitations. Firstly, it is a cross-sectional study based on the opinion and perception of women, and it is possible that information biases may exist. It should also be said that a comparative analysis of SARS-CoV-2 neonatal diagnosis with breastfeeding-related variables was not possible due to the low number of cases. In addition, we used a non-probabilistic sampling technique through social networks, which means that there may be information bias. It should be noted that although the sample was not random, the socio-demographic characteristics of the women coincide with the profile of pregnant women in Spain [38]. Despite these limitations, we believe that the results of this study are relevant.

## 5. Conclusions

Women and their desire to initiate and/or maintain breastfeeding have been significantly challenged during the SARS-CoV-2 pandemic. Thus, one in four women who participated in this study were offered formula milk during their hospital stay, and one in three women did not feel effectively supported in their doubts or difficulties, with the hospital setting being reported by two out of three women as the place where they did not feel supported. In addition, women who reported having experienced obstetric violence in their care were offered less support for breastfeeding, more formula feeding, less contact with support groups in the area, or more recommendations to abandon breastfeeding in order to be vaccinated against SARS-CoV-2. Therefore, it can be concluded that, in addition to all the variables that hinder breastfeeding support, maternal SARS-CoV-2 diagnosis and obstetric violence must now also be taken into consideration. Moreover, more research on the possible relationship between obstetric violence and breastfeeding is urgently needed.

## Figures and Tables

**Figure 1 ijerph-19-15737-f001:**
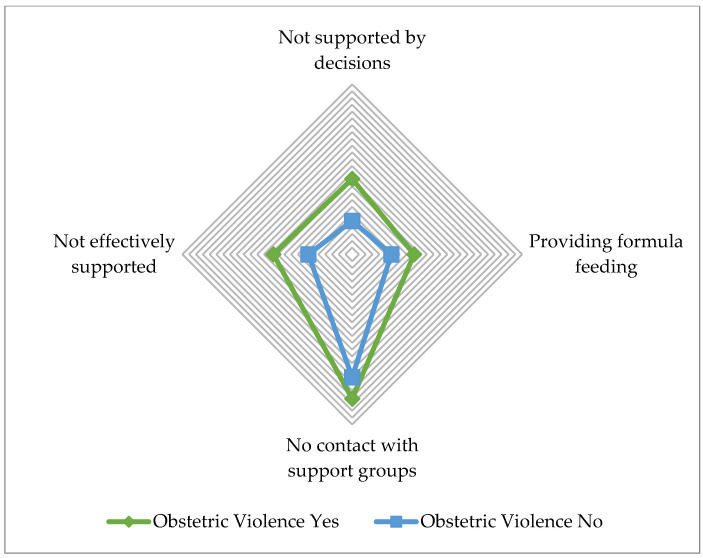
Comparative analysis of breastfeeding support and obstetric violence during the SARS-CoV-2 pandemic.

**Table 1 ijerph-19-15737-t001:** Sociodemographic and descriptive characteristics of participants (*n* = 6060).

	*n* (%)
**Occupation**	
Housewife	279 (4.6)
Student	55 (0.9)
Unemployed	579 (9.6)
Employed worker	4423 (73)
Self-employed	586 (9.7)
Other	138 (2.3)
**Level of education**	
Basic education	60 (1)
Secondary education	1264 (20.9)
University education	4736 (78.2)
**Social class**	
Lower	260 (4.3)
Middle	5664 (93.5)
Upper	136 (2.2)
**Race or ethnicity**	
Caucasian	5887 (97.1)
Romani	8 (0.1)
Black	11 (0.2)
Other	103 (1.7)
**Type of childbirth**	
Caesarean section	1309 (21.6)
Scheduled C-section	395 (30.18)
Urgent C-section	914 (69.82)
Instrumental birth	869 (14.3)
Vaginal birth	3882 (64.1)
**Type of healthcare**	
Private healthcare	584 (9.6)
Public healthcare	3572 (58.9)
Mixed healthcare	1904 (31.4)
**Maternal SARS-CoV-2**	
No	5797 (95.7)
Yes, during pregnancy	200 (3.3)
Yes, during childbirth	63 (1)
**Neonatal SARS-CoV-2 (*n* = 263)**	
No	255 (97)
Yes	8 (3)

**Table 2 ijerph-19-15737-t002:** Descriptive statistics of variables related to breastfeeding and obstetric violence (*n* = 6060).

	*n* (%)
**During the postpartum period, did you feel supported in your decisions about feeding and caring for your baby?**	
No	1590 (27.5)
Yes	3951 (68.3)
Do not know	246 (4.3)
**In which area did you feel unsupported? (*n* = 1590)**	
Primary care	381 (24.0)
Hospital	1046 (65.8)
Both in primary care and in hospital	163 (2.7)
**Offered formula feeding to help her rest or because she did not have enough milk**	
No	3920 (69.3)
Yes	1536 (27.2)
Do not know	198 (3.5)
**Were you put in contact with support groups in the area?**	
No	4315 (76.3)
Yes	1140 (20.1)
Do not know	204 (3.6)
**Did you feel effectively supported and helped to resolve doubts or difficulties?**	
No	1819 (32.5)
Yes	3783 (67.5)
**Were you advised to stop breastfeeding so that you could be vaccinated against COVID-19?**	
No	5034 (90.6)
Yes	88 (1.6)
Do not know	437 (7.9)
**Would you say you have experienced obstetric violence?**	
No	4030 (66.5)
Yes	1578 (26.0)
Do not know	452 (7.5)
**Obstetric violence is justified by the pandemic (*n* = 1578)**	
No	1376 (87.2)
Yes	57 (3.6)
Do not know	145 (9.2)

**Table 3 ijerph-19-15737-t003:** Comparative analysis of variables on breastfeeding and obstetric violence (*n* = 6060).

			Obstetric Violence								
	Total		No		Do Not Know		Yes				
	*n*	%	*n*	%	*n*	%	*n*	%	*X* ^2^	*df* ^1^	*p* ^2^
**During the postpartum period, did you feel supported in your decisions about feeding and caring for your baby?**											
No	1590	27.5	765	19.7	170	40.5	655	44.4	404.941	4	<0.001
Do not know	246	4.3	140	3.6	25	6.0	81	5.5			
Yes	3951	68.3	2988	76.8	225	53.6	738	50.1			
**In which area did you feel unsupported? (*n* = 1590)**											
Primary care	381	24.0	236	30.8	36	21.2	109	16.6	42.339	4	<0.001
Hospital	1046	65.8	459	60.0	121	71.2	466	71.1			
Both	163	10.3	70	9.2	13	7.6	80	12.2			
**Offered formula feeding to help her rest or because she did not have enough milk**											
No	3920	69.3	2802	73.8	249	60.3	869	60.3	119.467	4	<0.001
Do not know	98	3.5	123	3.2	27	6.5	48	3.3			
Yes	1536	27.2	487	23.0	137	33.2	525	36.4			
**Were you put in contact with support groups in the area?**											
No	4315	76.3	2745	72.3	342	82.6	1228	84.9	108.864	4	<0.001
Do not know	204	3.6	151	4.0	20	4.8	33	2.3			
Yes	1140	20.1	902	23.7	52	12.6	186	12.9			
**Did you feel effectively supported and helped to resolve doubts or difficulties?**											
No	1819	32.5	969	25.8	185	45.5	665	46.4	235.522	2	<0.001
Yes	3783	67.5	2793	74.2	222	54.5	768	53.6			
**Were you advised to stop breastfeeding so that you could be vaccinated against COVID-19?**											
No	5034	90.6	3441	92.2	347	85.5	1246	88.1	32.940	4	<0.001
Do not know	437	7.9	252	6.7	47	11.6	138	9.8			
Yes	88	1.6	46	1.2	12	3.0	30	2.1			

^1^ *df*, Degrees of Freedom; ^2^ chi-square test.

**Table 4 ijerph-19-15737-t004:** Comparative analysis of breastfeeding variables and the diagnosis of maternal SARS-CoV-2.

			Maternal SARS-CoV-2								
	Total		No		Yes, during Pregnancy		Yes, during Childbirth				
	*n*	%	*n*	%	*n*	%	*n*	%	*X* ^2^	*df* ^1^	*p* ^2^
**During the postpartum period, did you feel supported in your decisions about feeding and caring for your baby?**											
No	1590	27.5	1520	27.5	45	23.6	25	39.7	7.313	4	0.129
Do not know	246	4.3	238	4.3	7	3.7	1	1.6			
Yes	3951	68.3	3775	68.2	139	72.8	37	58.7			
**In which area did you feel unsupported? (*n* = 1590)**											
Primary care	381	24.0	362	23.8	16	35.6	3	12.0	6.697	4	0.153
Hospital	1046	65.8	999	65.7	27	60.0	20	80.0			
Both	163	10.3	159	10.5	2	4.4	2	8.0			
**Offered formula feeding to help her rest or because she did not have enough milk**											
No	3920	69.3	3747	69.3	137	73.3	36	58.1	5.358	4	0.252
Do not know	198	3.5	188	3.5	7	3.7	3	4.8			
Yes	1536	27.2	1470	27.2	43	23.0	23	37.1			
**Were you put in contact with support groups in the area?**											
No	4315	76.3	4126	76.3	140	74.5	49	79.0	0.895	4	0.925
Do not know	204	3.6	196	3.6	6	3.2	2	3.2			
Yes	1140	20.1	1087	20.1	42	22.3	11	17.7			
**Did you feel effectively supported and helped to resolve doubts or difficulties?**											
No	1819	32.2	1733	32.4	57	30.6	29	46.8	6.093	2	0.048
Yes	3783	67.5	3621	67.6	129	69.4	33	53.2			
**Were you advised to stop breastfeeding so that you could be vaccinated against COVID-19?**											
No	5034	90.6	4819	90.7	164	88.2	51	85.0	4.075	4	0.396
Do not know	437	7.9	410	7.7	19	10.2	8	13.3			
Yes	88	1.6	84	1.6	3	1.6	1	1.7			

^1^ *df*, Degrees of Freedom; ^2^ chi-square test.

## Data Availability

Data are available on request from the authors.

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
