# Peer review of "Breastfeeding and Obstetric Violence during the SARS-CoV-2 Pandemic in Spain: Maternal Perceptions"

_ijerph, 2022, doi:10.3390/ijerph192315737_

Round 1
Reviewer 1 Report
The authors analyze the perceptions of women in the maternity stage about obstetric violence and support for breastfeeding during the COVID-19 pandemic. For the study, they used data from 6060 women who gave birth in Spain between March 2020 and April 2021, obtained through an online questionnaire. After the corresponding statistical analysis, the authors found that women diagnosed with SARS-CoV-2 during delivery (p = 0.048), belonging to a low social class (p = 0.031), with secondary education (p = 0.029), or who suffered obstetric violence (p < 0.001) perceived less support, as well as the health care providers were less inclined to resolve doubts and difficulties about breastfeeding.
My comments are the following:
The literature review could be extended. For example, what have other authors found regarding your research topic, either in Spain or other countries?
“Thus, the aim of this study was to describe women's perceptions of obstetric violence related to breastfeeding support in Spain during the SARS-CoV-2 pandemic.” I think you analyzed, rather than described.
Provide more details of the applied instrument and its validation (e.g. Cronbach's alpha).
“Fig1. Obstetric violence and breastfeeding support during the SARS-CoV-2 pandemic.” Use a more descriptive title.
Figure 1 needs to be explained in more detail.
Mention future work in the Conclusions section.
Author Response
Dear Reviewer
Thank you for each and every one of your contributions. Undoubtedly, the final result of our manuscript is better with your contribution. Please find attached a table with the responses to each of your comments. I hope that our responses will be satisfactory to your approaches. We look forward to hearing from you if there are any further requirements.
Best regards.

Reviewer 2 Report
This is a very interesting article on a relevant topic and in the breastfeeding part, novel in its link to obstetric violence. After reviewing the sections in detail, I think it could benefit from some changes, especially in the discussion.
Introduction: Obstetric violence is mentioned, and the lack of consensus on its definition. Mention is made of its incorporation into legislation in Catalonia. The increase during the Covid19 pandemic is mentioned, specifically linking it to breastfeeding.
Suggestions for improvement:
1) It is my understanding that the legislation in Catalonia is the first in any region or country in Europe. I think this needs to be added.
2) I think that more development would be needed on the link between obstetric violence and breastfeeding, so that the justification of the research objectives would be clearer.
Methodology: The methodology used (cross-sectional study using an online survey) is explained in detail. Compliance with the ethical principles of the research is mentioned, as well as the statistical analysis.
Suggestions for improvement:
1) I believe that describing all the variables in this section would be necessary.
Results: The characteristics of the sample are described in detail. The variables related to breastfeeding are described according to the perception of obstetric violence, and according to the diagnosis of Covid infection of the pregnant woman. The tables and figure are easy to understand.
Suggestions for improvement:
1) None
Discussion: The results are discussed and compared with the literature on the subject. But I think that some of the interpretations of the results are debatable, and that more order could be followed in the discussion.
Suggestions for improvement:
1) The first paragraph states that the results show that breastfeeding has been compromised during the Covid pandemic by advice not to abandon breastfeeding. But the results indicate that this has occurred in 1.6% of women (2.1% of those who perceived obstetric violence). It seems to me that this is a very small percentage for such a categorical statement.
2) In the second paragraph it is stated that two thirds of the women indicated that they did not feel supported in the hospital. But the results indicate that it is two-thirds of the women who did not feel supported, i.e., two-thirds of the 27.5%, resulting in 17% of the total number of women. The interpretation of the number should not the same.
3) I may have missed some other such interpretation that does not match the results. I recommend reviewing the entire discussion section carefully.
4) Since the sample is not random, I think a summary of the characteristics of the sample and a comparison with the literature to indicate how it resembles or differs from the standard profile of pregnant women in Spain would be useful.
5) I think the discussion should follow the order of the variables according to the objective (perception of obstetric violence in breastfeeding). Comment on each variable that may be related to obstetric violence, and compare with the literature, then the significant ones according to whether or not the woman perceives obstetric violence with its related discussion and what is already of lack of specific research on breastfeeding as evidence of violence, or the lack of recognition of violence by women, and finally mention the lack of statistical relationship with the diagnosis of Covid in the pregnant woman.
6) In the comparison between the perception of violence or not and the breastfeeding variables, I believe that caution should be exercised in the interpretation of the data. Being a cross-sectional study, cause and effect can be confused. It is not clear to me whether the women who recognize obstetric violence are the ones who are more sensitized to it and, therefore feel less support, or whether it is the other way around, that those who have suffered the most are the ones who later, because of what has happened to them, realize that they have suffered obstetric violence. I do not know if I am making myself clear, if I am sensitized to gender violence, I will identify violence before or with actions that others will not realize, that is to say, a priori; but it can also be that if I have suffered gender violence, then I can realize a posteriori what has happened to me and identify it. The result is the same, but the cause and effect are reversed.
Conclusions: I recommend revising the conclusions and summary in light of the changes in the rest of the sections.
Author Response

(The authors gave the same response as above.)

Round 2
Reviewer 2 Report
Thank you for the revision you have made based on the reviewers' suggestions. I believe the changes have clarified and improved the article.